# Onboard Radio Frequency Interference as the Origin of Inter-Satellite Biases for Microwave Humidity Sounders

**Imke Hans \*, Martin Burgdorf and Stefan A. Buehler**

Meteorologisches Institut, Centrum für Erdsystem- und Nachhaltigkeitsforschung (CEN), Universität Hamburg, Bundesstrasse 55, 20146 Hamburg, Germany; martin.burgdorf@uni-hamburg.de (M.B.); stefan.buehler@uni-hamburg.de (S.A.B.)

\* Correspondence: imke.hans@uni-hamburg.de; Tel.: +49-40-42838-8121

**Abstract:** Understanding the causes of inter-satellite biases in climate data records from observations of the Earth is crucial for constructing a consistent time series of the essential climate variables. In this article, we analyse the strong scan- and time-dependent biases observed for the microwave humidity sounders on board the NOAA-16 and NOAA-19 satellites. We find compelling evidence that radio frequency interference (RFI) is the cause of the biases. We also devise a correction scheme for the raw count signals for the instruments to mitigate the effect of RFI. Our results show that the RFI-corrected, recalibrated data exhibit distinctly reduced biases and provide consistent time series.

**Keywords:** microwave humidity sounders; inter-satellite biases; radio frequency interference; consistent time series

---

## 1. Introduction

The existence of strong scan- and time-dependent inter-satellite biases is well-known in the microwave humidity sounders AMSU-B (Advanced Microwave Sounding Unit-B) and MHS (Microwave Humidity Sounder) onboard the NOAA-16 and NOAA-19 satellites compared to the MHS on NOAA-18 [1–4]. These biases prevent the construction of long, consistent time series as required for climate research. So far, a consistent correction of these biases based on a recalibration has not been possible because the origin of these biases was unknown. Instead, lookup tables with empirical calibration coefficients that change from month to month are employed as a makeshift correction method [5]. There have been further bias correction efforts (e.g., References [4,6]) for other instruments. The corrections usually aim at a reconciliation of the so far inconsistent sensors without finding the actual origin of the bias in the process of measurement or calibration. However, we now have strong evidence that radio frequency interference (RFI) from transmitters onboard the satellite is the origin for the observed biases for the AMSU-B on NOAA-16 and MHS on NOAA-19. We obtained this result from a dedicated analysis, which is presented in this article. Moreover, by devising a correction scheme and applying it within the calibration procedure, we were able to mitigate the effect of RFI and to reduce the biases distinctly for the sounding channels around the 183 GHz water vapour absorption line. This bias reduction increased the consistency of the two instruments among themselves and compared to NOAA-18.

In the context of microwave humidity sounders, the RFI from onboard transmitters was first observed for AMSU-B on NOAA-15. The electromagnetic signals from the antennas to send data from the satellite to

ground stations or the signals from the antennas acting as a search-and-rescue repeater caused interference effects within the AMSU-B instrument. The extra radiation entered the instrument through the main reflector, but the pickup of the signal occurred only at the back end of the instrument in the region of the video amplifiers/detectors [7]. Strong scan-dependent biases were observed, which differed between the channels.

Currently, the RFI from onboard transmitters is considered an important effect only for AMSU-B on NOAA-15 [7,8], which receives a correction scheme on the raw counts to mitigate this RFI effect. For the AMSU-B on NOAA-16 and on NOAA-17, the effect is reduced by improved shielding [7]. For NOAA-16, the remaining RFI-induced bias was below the noise level during the in-orbit verification phase and, hence, was considered negligible at that time [9]. For NOAA-17, a small correction was required, and a correction scheme was devised to be applied during calibration processing [9]. During the early years of NOAA-15, erratic changes in the RFI-induced bias pattern were observed, and the bias correction scheme was updated at first, but later, the updating efforts were stopped [9]. Consequently, it can be expected that NOAA-15 suffers from RFI in later years for which the correction scheme may be outdated and no longer suited to capture the RFI effect. For the other sensors, the impact of RFI might have changed as well. In Reference [3], the suspicion was formulated that AMSU-B on NOAA-16 may suffer from RFI because the gain decreased, and hence, the relative impact of RFI may have increased. Neither was this investigated further in Reference [3] nor were consequences formulated. In Reference [10], all other known calibration parameters were excluded as possible sources of bias using moon-intrusions as the analysis tool and RFI was suggested as a possible cause for the biases associated with NOAA-16. For MHS instruments, it seems that RFI has not been considered as a possible source of error, probably because of an improved instrumental design.

In this study, we analyse the scan- and time-dependent biases of AMSU-B on NOAA-16 and MHS on NOAA-19 versus MHS on NOAA-18, taking into account the instrumental behaviour, namely the evolution of the recorded raw signals for the calibration targets and the associated evolution of the gain. We show that RFI is the most plausible cause for the observed bias patterns, especially in light of the comparable patterns seen for AMSU-B on NOAA-15 and the known RFI problems of this instrument.

Moreover, we devise a correction scheme on the Earth view counts to reduce the RFI impact on the calibration. Using this correction scheme in the calibration procedure distinctly reduces the observed biases for NOAA-16 and NOAA-19 against NOAA-18 and provides a consistent time series.

The article is structured as follows. In Section 2, we provide a brief overview over the considered instruments and data. In Section 3, we explain the methods used for analysing the inter-satellite biases and we also present the results from the bias analysis, highlighting the impact of RFI on AMSU-B on NOAA-16 and on MHS on NOAA-19. In Section 4, we explain the deduction of the RFI correction scheme and we show the RFI-corrected time series for these instruments and the improved consistency among the instruments. Section 5 includes a discussion of the results, and Section 6 concludes the article.

## 2. Data

The considered instruments, AMSU-B and MHS, on the NOAA-16 and NOAA-19 satellites, respectively, are microwave humidity sounders that are very similar to each other, MHS being the successor instrument of AMSU-B. Here, we only briefly highlight their main characteristics. A more comprehensive description can be found in Reference [9]. The instruments are cross-track scanners with 90 Earth views. The viewing or scan angles covered by the 90 fields of view (FOVs) reach from $-49.5°$ for FOV 1 at the left-hand side of the swath in the flight direction to $49.5°$ for FOV 90 at the right-hand side of the swath, with $0°$ corresponding to the nadir view. The instruments have five radiometric channels at 89 GHz (channel 1), 150 GHz (channel 2, 157 GHz for MHS), $183 \pm 1$ GHz (channel 3), $183 \pm 3$ GHz (channel 4),

and 183 ± 7 GHz (channel 5, 190 GHz for MHS). In this article, we use the channel numbering from MHS, although the AMSU-B channels were originally labelled as channels 16–20. The three channels around the 183 GHz line (channels 3–5) are the sounding channels that are sensitive to water vapour absorption in the upper, middle, and lower tropospheres. Here, we focus mainly on these sounding channels because the applied bias analysis method and the RFI correction method are suited for channels with a small diurnal cycle impact only. Moreover, they are the most important for the derivation of humidity profiles.

The data used for the bias analysis were level 1c brightness temperatures processed from NOAA-CLASS level 1b data using the operational processing AAPP (ATOVS and AVHRR Preprocessing Package). We also used NOAA-CLASS level 1b data to analyse the time evolution of basic calibration quantities, as in Reference [11].

## 3. Bias Analysis

In this section, we present the methods used for the bias analysis and the corresponding results, indicating the RFI impact on NOAA-16 and NOAA-19.

### *3.1. Methods—Bias Analysis*

The strong biases for NOAA-16 and NOAA-19 have two important characteristics, namely the time and scan dependence. We analysed the bias on the basis of global monthly means of the individual satellite data, with averaged ascending and descending passes. As a reference, we used MHS on NOAA-18. Any bias is, therefore, meant as a SENSOR–MHS on NOAA-18.

#### 3.1.1. Time Dependence

In order to analyse the time dependence of the bias, we also averaged the overall scan angles, i.e., all fields of view (FOVs). The resulting bias is plotted as a function of time (see Figure 1, left panels). The bias evolution information was then evaluated together with the evolution of the raw counts from the calibration views, which are the signals recorded when the instrument looks at the hot and at the cold calibration targets. The temperatures of these targets are known. The cold target, being the cosmic microwave background, is stable, and the temperature of the warm target is constantly measured. Therefore, any evolution of the recorded raw signals can be analysed and classified as an instrumental effect. Looking at both the bias and the evolution of the raw signals or of the related gain (see Figure 1, right panels) together helps to identify the causes of a bias.

#### 3.1.2. Scan Dependence

We analysed the scan dependence of the bias by plotting the global monthly mean bias as a function of the viewing angle for each month and year. This gave an overview of the observed patterns of the scan dependence and of the evolution of these patterns over time.

### *3.2. Results—Bias Analysis*

In this section, we present the results from the bias analysis, revealing the impact of RFI on NOAA-16 and NOAA-19.

#### 3.2.1. RFI Impact on AMSU-B on NOAA-16

The bias of NOAA-16 showed a strong time dependence (see the left panels in Figure 1, blue curve). In the early years, it was below 0.5 K for channel 3, whereas in later years, it even exceeded 1.5 K. For channels 4 and 5, the absolute value of the bias was even larger. The time dependence of the bias was highly correlated with the evolution of the raw counts from the warm and cold target views and the

deduced gain, as shown in the right panels of Figure 1. Note the overall decrease in the gain and the overall bias increase. Note also the distinct pattern in 2008–2009 and 2009–2010 related to the change in the solar beta angle as the satellite approached a local equator crossing time (LECT) of 18:00 [6]. This pattern in the gain translates into the evolution of the observed bias. We conclude that the observed bias evolution relates to changes in the raw signals and the corresponding gain, which are not the cause of the bias by themselves but may act as catalysts for the RFI effect to become significant. This close but indirect relation of the decreasing gain and raw signals with the increasing bias was observed for AMSU-B on NOAA-15 too (see Figure 1, grey curves). For this instrument, the issue of RFI impact is well-known. Hence, after observing the same evolution for NOAA-16 and knowing that NOAA-16 was also slightly affected by RFI at the beginning of the mission though below the noise level, we deduced that NOAA-16 is severely affected by RFI in its later years when the raw signals and the gain decrease (as suggested in Reference [3], though without further investigation). As a consequence of the decreased signals from the calibration views and Earth views, the relative impact of the extra radiation due to RFI, which is picked up at the back end of the receiver only [7] and added to the recorded Earth view counts $C_E$, is increased and, hence, introduces an increasing bias over time. This catalysing effect can be seen from the basic two-point calibration equation used for the microwave (MW) sounders, which, using the Rayleigh–Jeans approximation that is valid for the typical brightness temperatures of Earth at MW frequencies, are

$$T_B = T_w + \frac{T_w - T_c}{C_w - C_c} \cdot (C_E - C_w) \tag{1}$$

$$= T_w - \frac{1}{G} \cdot C_w + \frac{1}{G} \cdot C_E \tag{2}$$

$$\text{with} \quad G = \frac{C_w - C_c}{T_w - T_c}, \tag{3}$$

where $T_B$ is the measured brightness temperature of the Earth pixel, $T_w$ and $T_c$ are the known temperatures of the warm and cold targets, $C_w$ and $C_c$ are the raw signal counts recorded at the warm and cold target views, $C_E$ denotes the Earth counts, and $G$ is the gain.

If instead of $C_E$, a contaminated $C'_E = C_E + C_{\text{RFI}}$ is recorded and processed in Equation (3), then a decreasing gain $G$ with reduced raw signals $C_E$ and $C_w$ will induce an increasing weight of $C_{\text{RFI}}$ in

$$T_B = T_w - \frac{1}{G} \cdot C_w + \frac{1}{G} \cdot C_E + \frac{1}{G} \cdot C_{\text{RFI}}. \tag{4}$$

Therefore, the RFI may increasingly contaminate $T_B$ if the gain and the raw signals decrease over time, while the external RFI effect $C_{\text{RFI}}$ is constant.

The second argument for RFI being the origin of the strong biases of NOAA-16 relates to the scan dependence of the bias. Figure 2b shows the global monthly mean bias as a function of the scan angle (or FOV) for October for the years 2005–2011 for channel 3 of AMSU-B on NOAA-16. October 2005 shows the smallest bias and was, therefore, chosen as the reference month in the later RFI correction. The overall bias increase over the years is clearly visible in Figure 2b. Most importantly, however, the bias pattern is almost stable over the years. The smoothly varying scan dependence varies slightly over time, with a stronger increase for FOVs near nadir (scan angle of zero) than for FOVs at the left edge. The zigzag pattern, however, is very stable throughout the years. These observations, again, in light of similarities to NOAA-15 (the time dependence in Figure 1 and the scan dependence in Figure 2a), clearly suggest an impact of RFI.

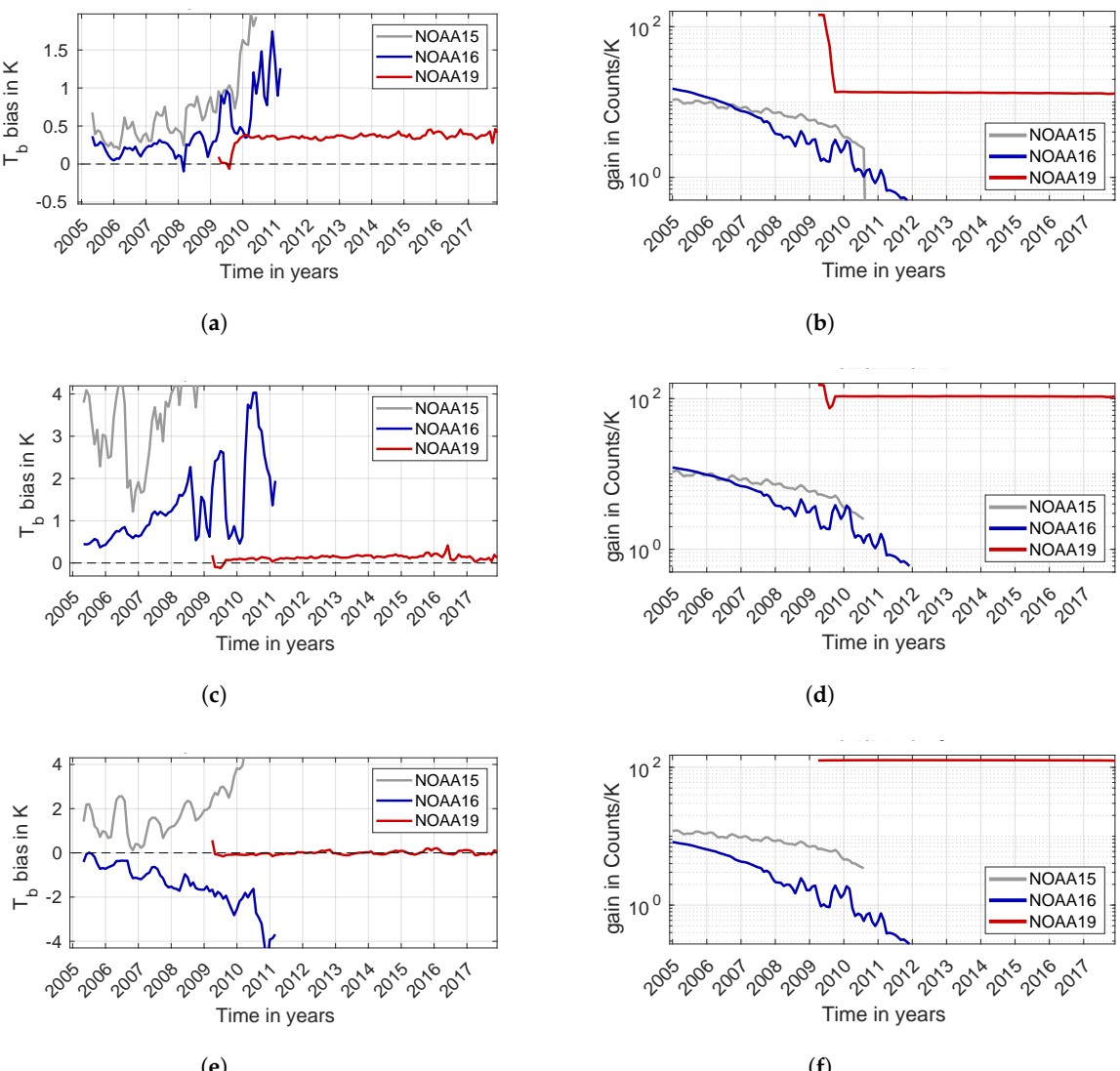

**Figure 1.** Left panels: The inter-satellite biases of AMSU-B (Advanced Microwave Sounding Unit-B) on NOAA-15 and NOAA-16 and of MHS (Microwave Humidity Sounder) on NOAA-19 versus the MHS on NOAA-18 for the three sounding channels. Right panels: The gain evolution of the channels in the same period. Note the time evolution of the biases compared to the evolution of the gain, especially the strong decrease of the gain for all three sounding channels (3–5) for AMSU-B and the sudden decrease in MHS for channels 3 and 4 on NOAA-19 together with the increasing absolute value of the bias. (**a**) Channel 3 bias; (**b**) Channel 3 gain; (**c**) Channel 4 bias; (**d**) Channel 4 gain; (**e**) Channel 5 bias; (**f**) Channel 5 gain.

To correct for this RFI impact, we derived the correction scheme explained in Section 4.1. This correction scheme is based on the bias analysis, as displayed in Figure 3. Figure 3 shows the scan-dependent bias for all months of 2007 and 2010, respectively. The overall increase and the stability of the zigzag pattern are clearly visible. The different increases in the smoothly varying scan dependence for the left-hand-side FOVs compared to the nadir FOVs and the right-hand-side FOVs are also visible. Smooth variations over time required a continuously varying correction scheme (see Section 4.1) at risk of

erroneously removing the diurnal cycle effects that translate to seasonal cycle effects in the monthly means and that change as the satellites drift. However, from Figure 3 (and the corresponding ones for all other years), we can see that there is no seasonal effect, but the dominating change over the year corresponds to an increased bias towards the end of the year. This coincides with the overall gain decrease over the lifetime of NOAA-16.

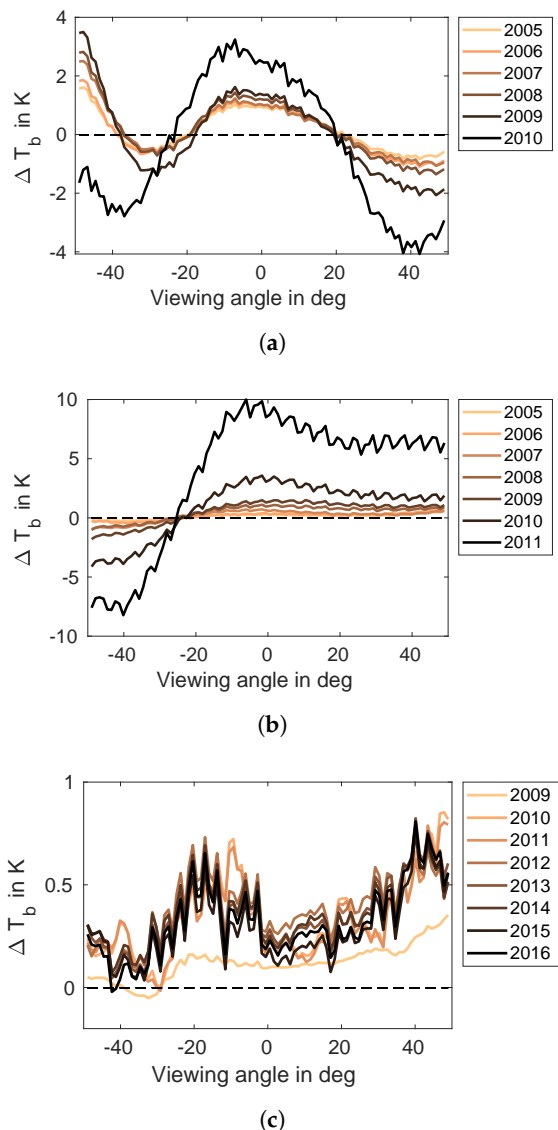

**Figure 2.** The scan-dependent bias for NOAA-15–NOAA-18, NOAA-16–NOAA-18 and NOAA-19–NOAA-18 in channel 3: The plot shows the global monthly means for August (NOAA-15) and October (NOAA-16) for the years 2005 to 2010/2011 and for April (NOAA-19) for the years 2009 to 2016. The plot shows the uncorrected NOAA-15 data, i.e., the RFI correction in AAPP is also switched off. (**a**) NOAA-15; (**b**) NOAA-16; (**c**) NOAA-19.

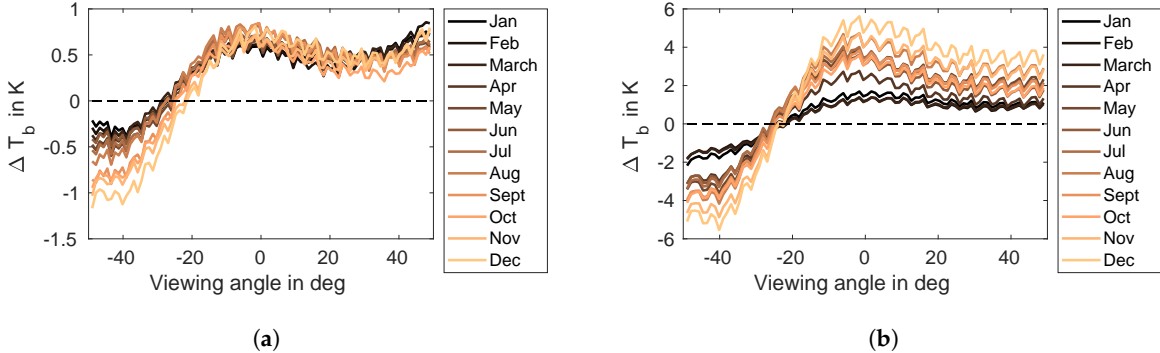

**Figure 3.** AMSU-B on NOAA-16: The scan-dependent bias for NOAA-16–NOAA-18 in channel 3. The plots show global monthly means for the years 2007 and 2010 for all months. No distinct seasonal pattern is visible. The bias increases over the course of the year as the gain decreases. (**a**) NOAA-16, Channel 3, 2007; (**b**) NOAA-16, Channel 3, 2010.

### 3.2.2. RFI Impact on MHS on NOAA-19

For MHS on NOAA-19, we carried out the same analysis as for NOAA-16. Comparing the bias to the evolution of the raw signals and the gain, we again observed a close relation, which is displayed in Figure 1 (red curves). When the gain suddenly dropped for channel 3 in summer 2009, the bias emerged. The gain then remained stable at the decreased level, and the bias remained stable at a constant value of about 0.4 K in channel 3 (see Figure 1a,b, red curves). The situation was similar for channel 4, although with a weaker drop in gain and an emerging bias (see Figure 1c,d, red curves). Channel 5, however, showed neither a decrease in raw signals and gain nor a bias in the brightness temperature (see Figure 1e,f, red curves). Hence, any RFI impact remained small for this channel, since the raw signals did not decrease, and hence, the relative impact of RFI did not increase. In conclusion, we observed the same qualitative behaviour of raw signals and gain on the one side and bias on the other side. Again, the catalyst role of the change in the gain was visible as in the case of NOAA-16 (see Section 3.2.1).

The scan dependence of the bias in channel 3 is shown in Figure 2c for April for the years 2009 to 2016. In April 2009, the zigzag pattern was the weakest, and therefore, we chose this month as the reference month for the later RFI correction. The bias seen in April 2009 is due to the antenna pattern correction (APC) of NOAA-18, which is the APC of NOAA-15 and is not suited for an MHS instrument [12]. This bias of a different origin than RFI was not considered within our RFI correction scheme (Section 4.1) but was absorbed in the reference month and, hence, was not overcorrected. For the later years, a clear zigzag pattern over the viewing angle was visible. Moreover, two different periods can be distinguished until 2011 (the orange curves in Figure 2c) and after 2011 (the brown to black curves), when the pattern changed.

Within these periods, before and after 2011, the RFI pattern did not change, and we assumed that a constant RFI correction is appropriate (see Section 4.1). The risk of erroneously removing the diurnal/seasonal cycle effects was small due to this constant correction. Visible changes in the bias because of the diurnal/seasonal cycle effects emerged in later years when the satellites drifted apart. Figure 4 shows the scan-dependent bias for the years 2012 and 2016. In 2012, no seasonal effect was visible, whereas for 2016, the months of the DJF (December-January-February) period showed a similar bias, different from the JJA (June-July-August) period. The RFI zigzag pattern is the same, however. Hence, a correction scheme on the Earth counts that taking into account this RFI pattern is a promising approach. The results are presented in Section 4.2.

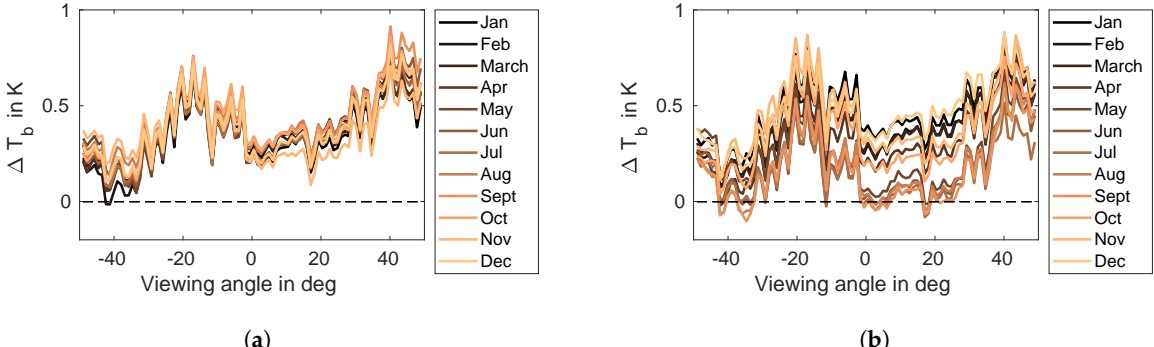

**Figure 4.** MHS on NOAA-19: The scan-dependent bias for NOAA-19–NOAA-18 in channel 3. The plots show global monthly means for the years 2012 and 2016 for all months. Note the emergence of a seasonal pattern in the bias as the two satellites drift apart. In 2016, the scan-dependent bias is very similar in the DJF (December-January-February) period of the year and differs from the JJA (June-July-August) period. (**a**) NOAA-19, Channel 3, 2012; (**b**) NOAA-19, Channel 3, 2016.

## 4. RFI Correction

In order to mitigate the strong impact of RFI presented in the previous section, Section 3.2, we developed an RFI-correction scheme. In this section, we present the method for deriving the RFI correction scheme and we show the corresponding results, namely bias reduction, when applying the RFI correction scheme to NOAA-16 and NOAA-19.

### 4.1. Methods—Deriving the RFI Correction Scheme

The RFI correction scheme is based on the analysis of the scan-dependent bias and its evolution over the years. The overall idea is to use the RFI bias in the brightness temperature and to convert it to counts using the gain of the corresponding period. This is based on the two-point calibration equation, Equation (3), from which we have

$$T_B = T_w - \frac{1}{G} \cdot C_w + \frac{1}{G} \cdot C_E. \tag{5}$$

If we had $C_E' = C_E + C_{\mathrm{RFI}}$ instead of $C_E$ in Equation (5), then there would be a change in $T_B$:

$$\mathrm{d}T_B = \frac{1}{G} \cdot C_{\mathrm{RFI}}. \tag{6}$$

The approach of our correction scheme is to extract this $C_{\mathrm{RFI}}$ from an observed bias $\Delta T_B = T_B(\mathrm{sen.}) - T_B(\mathrm{ref.})$ for a sensor and a reference. The obtained counts correspond to the extra amount of recorded signals due to an RFI that we will have to subtract from the recorded $C_E'$. This count value $C_{\mathrm{RFI}}$ (rounded to the nearest integer) is computed for each FOV such that we can finally provide a correction scheme for the Earth counts of each Earth view.

The details are described in the following paragraphs. An overview of the development of the RFI correction is given in Figure 5.

An important aspect of the derivation of the correction is the definition of the RFI bias, which means the portion of the bias that can be assigned to RFI (see the upper parts of Figure 5). From the global monthly mean biases used to investigate the scan dependence, the information of how much the sensors and MHS on NOAA-18 (reference $R$) differ on average is obtained. However, this difference includes

several aspects. First, it includes the difference due to the RFI impact. Hence, for the monthly mean bias we write

$$\Delta T_B = T_B' - T_R \tag{7}$$
$$= T_B + \Delta T_{B,RFI} - T_R, \tag{8}$$

if the sensor is affected by RFI ($'$ is used again to denote the contamination). However, the difference may also include instrumental issues not related to RFI. These are assumed to be constant over time and to, therefore, cancel out in the following steps. This is a necessary assumption at this point. Moreover, the difference includes effects due to the sampling of different phases of the diurnal cycle. These effects are smaller for the sounding channels than for the surface channels, and therefore, this method is only used for the sounding channels. From the listed contributions to the overall inter-satellite bias, the RFI contribution is extracted as follows. A month of a specific year is chosen where the RFI pattern is weakest (see Figure 2b,c for 2005 for NOAA-16 and 2009 for NOAA-19, respectively). This month $m_0$ is then defined as RFI-free $\Delta T_{B,RFI}(m_0) = 0$ (which is not necessarily true, as there might be a small impact still). Computing the bias change from the reference month $m_0$ to any month $m$ gives

$$\delta \Delta T_B = \Delta T_B(m) - \Delta T_B(m_0) \tag{9}$$
$$= T_B(m) + \Delta T_{B,RFI}(m) - T_R(m) - (T_B(m_0) + \Delta T_{B,RFI}(m_0) - T_R(m_0)) \tag{10}$$
$$= (T_B(m) - T_R(m)) - (T_B(m_0) - T_R(m_0)) + \Delta T_{B,RFI}(m). \tag{11}$$

At this point, it is assumed that all other instrumental issues causing biases are constant in time. Our second assumption is that possible changes in the monthly mean differences due to variations in the diurnal cycle from month $m_0$ to month $m$ are negligible. (The validity of this assumption is ensured in the instrument-specific definition of the correction; see next paragraphs.) With the two assumptions, the expressions in parentheses in Equation (11) cancel to yield

$$\delta \Delta T_B = \Delta T_{B,RFI}(m). \tag{12}$$

Any change $\delta \Delta T_B$ in the bias between the sensor and the reference with respect to this reference month $m_0$ is, therefore, defined as the RFI bias:

$$\Delta T_{B,RFI}(m) = \delta \Delta T_B = \Delta T_B(m) - \Delta T_B(m_0) \tag{13}$$
$$\Rightarrow \quad C_{RFI}(m) = G(m) \cdot \Delta T_{B,RFI}(m). \tag{14}$$

In the worst case, this $\Delta T_{B,RFI}(m)$ would include a contribution due to the diurnal cycle, and we would erroneously remove the diurnal cycle differences when later applying the correction $-C_{RFI}(m)$. Hence, we need to be more careful and specific in the definition of the correction based on this RFI bias for the individual instruments in order to keep the contribution from the diurnal cycle as small as possible.

The instrument-specific definition of the correction is depicted in the lower part of Figure 5. The deduction of the correction started with studying the monthly mean bias of the instrument and the reference $\Delta T_B$ over the duration of the mission. By studying the plots of the monthly mean bias $\Delta T_B$ like Figure 3 (for NOAA-16) and Figure 4 (for NOAA-19) for all years of the respective mission, we were able to deduce whether the RFI pattern remained constant or changed over time.

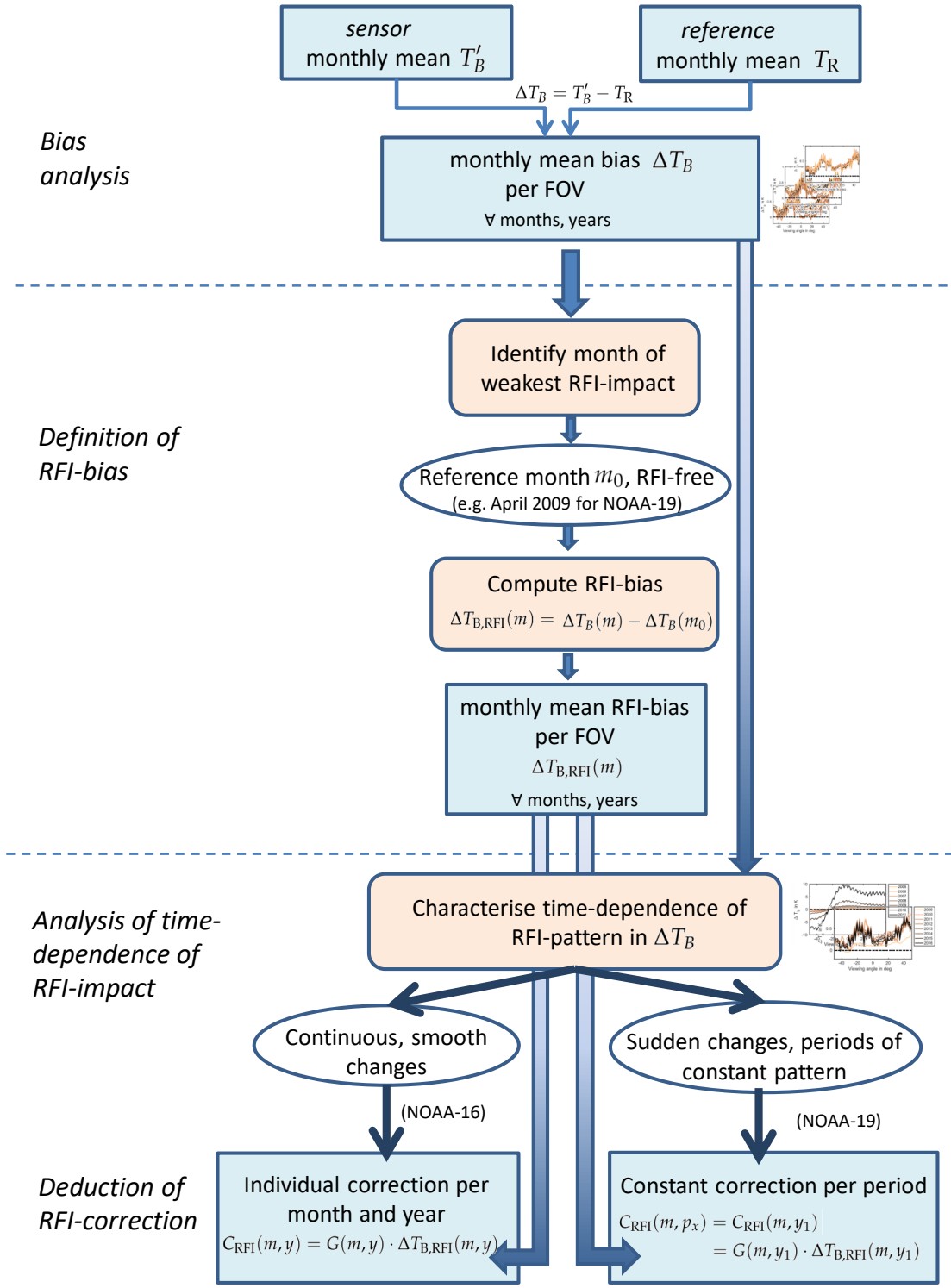

**Figure 5.** A flowchart visualising the steps towards the radio frequency interference (RFI) correction: The symbols are explained in the text.

For MHS on NOAA-19, we identified distinct periods of constant RFI impact (see Section 3.2.2). Hence, we only needed a constant correction per period. Each of these corrections was obtained from the RFI bias (bias change with respect to reference month April 2009) for all months of the first year in the specific period. We chose April 2009, since the RFI impact in this month was weak and the unstable phase of summer 2009 only started later. The correction (per month) of the first year $y_1$ in the specific period $p_x$ was then used as a correction for the respective months of all other years in the period. Hence, for all years of period $p_x$

$$C_{\text{RFI}}(m, p_x) = C_{\text{RFI}}(m, y_1) = G(m, y_1) \cdot \Delta T_{\text{B,RFI}}(m, y_1) \tag{15}$$

where month $m$ is Jan, Feb, ... Dec, and $G(m, y_1)$ denotes the median gain of the sensor in month $m$ of the first year $y_1$ in the respective period $p_x$. See Table 1 and Figure 6 for the applied correction scheme on Earth view counts. As the analysis of the scan-dependent bias has shown, the impact of the diurnal cycle effects (that translate to seasonal cycle effects in the monthly means) was smaller than the RFI impact and only emerged for the later years when NOAA-19 and the reference NOAA-18 drifted apart. Since the RFI-correction is based on the first year of each period, even the correction for the third period $p_3$ from December 2011 to 2017 was based on the relatively early year 2012, when the impact of the diurnal cycle effects was small. Hence, the applied derivation to obtain the RFI correction ensured a reduced contribution from the diurnal cycle effects.

For NOAA-16, no clear periods of constant RFI pattern were identified (see Section 3.2.1). Rather, the pattern changed continuously over time to a further increased positive bias for FOVs around nadir and a further increased negative bias towards FOVs at the left edge of the scan (see Section 3.2.1), superimposed with a general increase in the absolute value of the bias over time. This behaviour does not allow for a clear definition of periods suitable for a constant correction. Hence, we defined the RFI bias as the difference in the bias of each individual month with respect to the reference month, October 2005. Consequently, each month and year received an individual correction:

$$C_{\text{RFI}}(m, y) = G(m, y) \cdot \Delta T_{\text{B,RFI}}(m, y). \tag{16}$$

See Table 2 and Figure 7 for the applied correction scheme on Earth view counts. This approach of individual, monthly RFI corrections, in general, overcorrected the differences due to diurnal and seasonal cycle effects, which should not be removed. However, the impact from RFI was much larger than these natural effects. In the bias analysis, we observed that the bias change did not follow a seasonal pattern but followed only the evolution of the increasing RFI impact as a consequence of the decreasing gain (see Section 3.2.1). Hence, in light of the dominating RFI impact, the effect of neglecting this possible overcorrection is small. It grows over time as the gain decreases because any bad correction value on the count level has an increasing impact when the gain decreases. This was accounted for in our uncertainty estimates.

**Table 1.** The NOAA-19–MHS RFI correction scheme assignment over the years: Each year belonging to a specific identified period $p_x$ (left column) received a correction from a specific year (columns for each channel). Note that channel 5 did not show a significant RFI impact and was, therefore, not corrected. Note also that December 2011 already showed the pattern of 2012 and, hence, received the corresponding correction from December 2012.

| NOAA-19<br>Year ...<br>Receives Correction From ... | Channel 3 | Channel 4 | Channel 5 |
|---|---|---|---|
| 2009 ($p_1$) | 2009 | 2009 | n/a |
| 2010 ($p_2$) | 2010 | 2010 | n/a |
| 2011 ($p_2$) | 2010 | 2010 | n/a |
| 2011-12 ($p_3$) | 2012-12 | 2012-12 | n/a |
| 2012 ($p_3$) | 2012 | 2012 | n/a |
| 2013–2017 ($p_3$) | 2012 | 2012 | n/a |

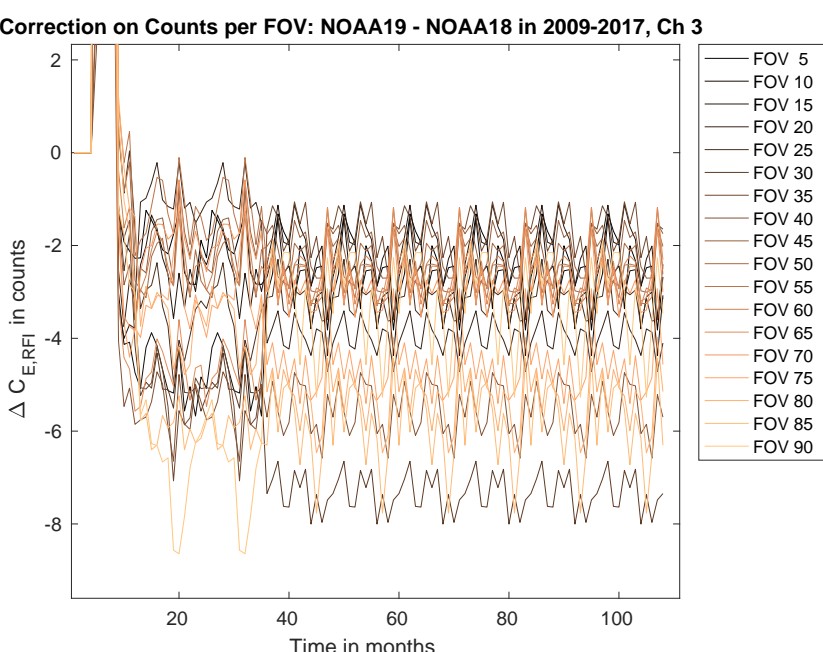

**Figure 6.** The NOAA-19-MHS RFI correction from January 2009 to December 2017 (months 1 to 108): The figure shows the correction only for every fifth field of view (FOV) for better readability. Clear periods of constant bias pattern were identified. These are reflected in the correction shown in this figure. Note the change from November 2011 (month 35) to December 2011 (month 36).

**Table 2.** The NOAA-16-AMSU-B RFI-correction scheme assignment over the years: Each year (left column) receives a correction from a specific year (columns for each channel). Note that all years before 2005 can only receive the correction from 2005 since there is no overlap with NOAA-18 to derive a more suited scheme.

| NOAA-16 Year Receives Correction From | Channel 3 | Channel 4 | Channel 5 |
|---|---|---|---|
| 2000–2005 | 2005 | 2005 | 2005 |
| 2006 | 2006 | 2006 | 2006 |
| 2007 | 2007 | 2007 | 2007 |
| ⋮ | ⋮ | ⋮ | ⋮ |
| 2012 | 2012 | 2012 | 2012 |

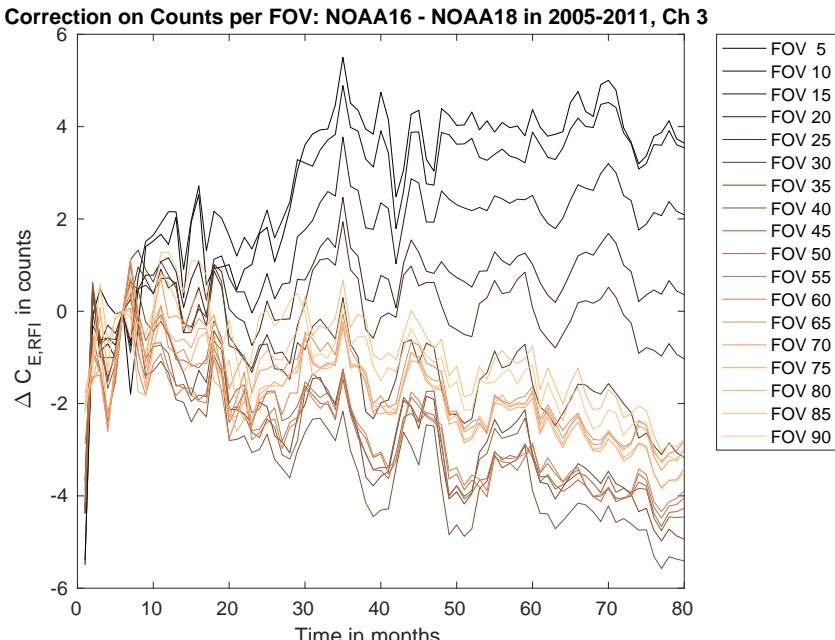

**Figure 7.** The NOAA-16-AMSU-B RFI correction from January 2005 to August 2011 (months 1 to 80): The figure shows the correction only for every fifth FOV for better readability. It is clearly visible that the correction derived from the underlying bias varies smoothly over time in a different way for the left-hand-side FOVs (5–45) than for the right-hand-side FOVs (50–90).

Having defined the instrument-specific RFI correction, it can be applied in our calibration processing. The correction is defined as

$$C_{\text{RFI, corr}} = -C_{\text{RFI}} \tag{17}$$

and rounded to the nearest integer. It is added to the contaminated Earth counts for each individual pixel (single measurement within a scan line within an orbit) as

$$C_{\text{E,corr}}(\text{pixel}) = C'_E(\text{pixel}) + C_{\text{RFI, corr}}(\text{FOV, month}) \tag{18}$$

where $C_{\text{RFI,corr}}$ (FOV, month) contains the correction for each FOV and the corresponding month the pixel belongs to.

The uncertainty estimate on our RFI correction scheme is twofold. First, the imperfection of the reference month is accounted for, which may be contaminated by RFI already (this is indeed probable). As the uncertainty estimate, the standard deviation over the FOVs of the reference month is used. This may provide a first indication of the initial RFI contamination. For NOAA-16, for channels 3–5, values of 0.20, 0.27, and 0.49 K were respectively obtained. For NOAA-19, for channels 3 and 4, values of 0.08 and 0.11 K were obtained, respectively. On top of that, a further uncertainty estimate related to a possible erroneous elimination of diurnal cycle effects is conducted. The standard deviation of the correction in counts over the months and FOVs is used. This uncertainty in counts then scales with the gain to an uncertainty estimate of the brightness temperature. This uncertainty estimate was about 0.1 K for NOAA-19, and it varied for NOAA-16 from 0.1 K in the early years to 1 K in 2011, which was correlated to the decrease in gain.

### 4.2. Results—RFI-cCrrected NOAA-16 and NOAA-19

The RFI correction scheme for NOAA-16 and NOAA-19 provided a value to be added to the Earth counts for each FOV, channel, and period of constant RFI. This extra proportion of counts was added to the Earth counts before any quality checks were executed in our processing chain for the FCDR (fundamental climate data record) generated within the Horizon 2020 project FIDUCEO (Fidelity and uncertainty in climate data records from earth observation) [12]. The calibration was then executed using these modified Earth counts in the measurement equation. Hence, the whole archive of the level 1b NOAA-16 and NOAA-19 data was recalibrated from the raw data, finally providing an RFI-corrected record of the brightness temperature.

The performance of this RFI correction is analysed in the following text. Figure 8 shows the global monthly mean bias as a function of time for the operational data (AAPP processed; dashed lines in Figure 8) and the RFI-corrected data (FIDUCEO FCDR processing; solid lines) for channels 3–5, together with the estimated uncertainties. A distinct improvement in terms of the stability and consistency to NOAA-18 was achieved. The strong bias increases for NOAA-16 (blue curves) were mitigated. The uncertainty of the applied correction indicates that the corrected data agree with NOAA-18. Small differences occurred because of an unsuited antenna pattern correction used for NOAA-18 in AAPP [12] and because of the different LECT of the satellites. The difference in LECT causes an overall offset of the instruments that varies slightly over time as the satellites drift, and it also induces the seasonal changes with an increasing amplitude for the times with increasingly different LECT. These remaining small differences also show that the RFI-correction does not erroneously remove the diurnal cycle effects. Overall, a stable improvement with respect to the uncorrected data was visible.

The correction for MHS on NOAA-19 (red curves in Figure 8) was also successful. Channel 3 agrees with NOAA-18 for the uncertainties (Figure 8a). The large stable offset against NOAA-18 was reduced. As for NOAA-16, the remaining small difference was due to the wrong antenna pattern correction being applied on NOAA-18 in the AAPP-processed data. This improved in our FCDR production presented in Reference [12]. The uncertainties due to common effects displayed as a shaded region still include the AAPP data at their upper bound. This is not an intention of the design of the RFI correction but a result of the combination of all the estimated uncertainties due to common effects (see References [12,13]). The RFI impact was much weaker for channel 4 (Figure 8b), and therefore, no obvious improvement was visible, as is the case for channel 3. In summer 2009, the AAPP processing provided a slightly smaller bias than the RFI-corrected data. This is also visible for channel 3. During this period, NOAA-19 was very unstable, with several interventions by the instrument control on the ground [14], such that the RFI

correction scheme was not expected to mitigate the RFI effect correctly. This period is not part of the FCDR presented in Reference [12]. Note that channel 5 is not RFI-corrected, and therefore, our processed data agree with the AAPP data (hence, the AAPP curve is hidden behind our data in Figure 8c). Finally, it should be noted that the applied RFI correction did not erroneously remove the diurnal/seasonal cycle effects, as we saw an increased amplitude for the seasonal cycle towards later years.

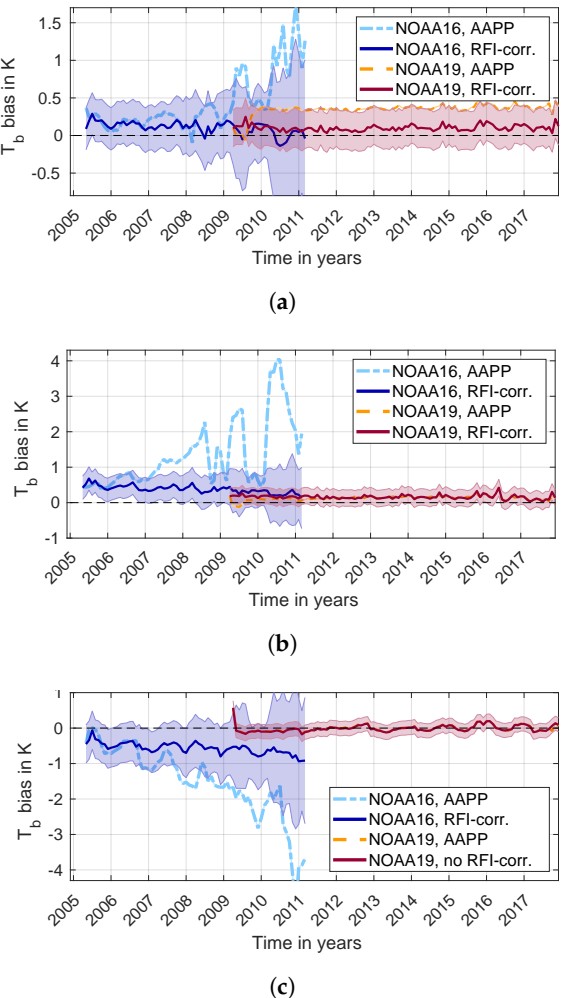

**Figure 8.** A comparison of the inter-satellite biases of MHS on NOAA-19 and AMSU-B on NOAA-16 against MHS on NOAA-18 in the operational AAPP data and in our RFI-corrected data as a function of time: The data show global monthly means averaged over all FOVs. The shaded regions denote the uncertainty due to *common* effects in the calibration. They include the propagated uncertainty estimate for the RFI correction as well as the uncertainties from various calibration parameters. They do not include noise effects. See Reference [12] for details on the uncertainties from the calibration parameters. (**a**) Channel 3; (**b**) Channel 4; (**c**) Channel 5.

## 5. Discussion

Our analysis of the global monthly mean biases of AMSU-B on NOAA-16 and MHS on NOAA-19 in terms of the time and scan dependence of the bias has revealed that both instruments are most likely affected by RFI in their sounding channels. For NOAA-16, RFI was judged as being negligible at the

start of the mission after the shielding had been improved compared to NOAA-15 [7]. There has been no further comment on the possible RFI on NOAA-16 in later years, except for the suggestion in Reference [3], where the possibility was formulated for a growing RFI impact as the gain decreases. For MHS, there has been no mention of RFI, as far as we know. In this regard, our analysis shows a very important result, first, because it reveals the origin of the inter-satellite biases that prevents the construction of a stable, consistent long time series. Recent attempts to produce a consistent long time series by using interpolation as an intercalibration method still results in biases of, for example, up to 4 K for channel 5 of NOAA-16 [5]. Having shown the impact of RFI and having provided a dedicated correction, we can improve the consistency of NOAA-16 and NOAA-19 with NOAA-18. Second, our result may also be of interest for the remaining MHS instruments in orbit. In the event of a strong gain decrease, an increased relative RFI impact may occur and lead to inter-satellite biases. Weak but stable zigzag patterns can be observed already for the other MHS on Metop-A and -B, too. In this regard, a controlled gain decrease for an instrument close to the end of its lifetime would be an interesting way to validate our results.

The RFI correction on the NOAA-16 and NOAA-19 was successful and increased the stability and consistency between the instruments. Our RFI correction scheme provided an individual correction for each FOV, thus respecting the high variability of the pattern between the FOVs. This is an improvement compared to the correction scheme that was developed during the in-orbit verification phase of NOAA-15 [7,8], where the correction was interpolated between FOVs. During this verification phase, the RFI on NOAA-15 could be investigated in a more direct way by switching the transmitters off and on and relating the correction to the transmitter power. This type of investigation was impossible for us because of the retrospective nature of our study on the long existing data from the NOAA-16 and NOAA-19 satellites. However, this retrospective analysis brought about the advantage that we had more data, allowing the RFI pattern to be extracted more precisely, and we were able to get an overview of the time evolution, which was unpredictable and erratic and would, therefore, require continuous monitoring and the adjustment of an operational correction.

Although our RFI correction scheme was successful at improving the overall consistency of the considered instruments, there were imperfections, which are explained in the following text. The correction scheme relies on a reference month that is assumed to be free of RFI contamination. This is a necessary assumption for this correction method, but it neglects the fact that the reference month is probably subject to an RFI impact, too. This leads to an incorrect RFI correction. Thus, we accounted for that in our uncertainty estimates. This requirement of a (mostly) uncontaminated reference month made it difficult to apply this method to the other RFI-affected AMSU-Bs on NOAA-15 and NOAA-17. For these instruments, it is not possible to define a clean reference month. One has to find individual solutions to find a satisfactory correction. We attempted to find such solutions for the FIDUCEO FCDR [12].

Since we use monthly means, our inter-satellite comparison method is subject to diurnal cycle effects, as mentioned before. This might lead to an erroneous elimination of diurnal/seasonal cycle effects, which we accounted for in our uncertainty estimate. A different approach free from diurnal cycle effects is to use simultaneous all angle collocations (SAACs), as in Reference [3]. The problem with this approach might be that because of less data, the patterns will not emerge as prominent as in the monthly means. In Reference [3], very similar patterns to those presented here were seen for NOAA-16. However, to collect more data, the SAAC method has to average the data from over many years, which does not give a clear picture of the evolution of the RFI impact. In this way, it could also lead to overcorrections. Nonetheless, it is an interesting possibility for cross-checking the RFI correction found.

Another shortcoming of the RFI correction scheme presented here is that the calibration views for the warm and cold targets that are not accessible. These views may also be contaminated by RFI and, hence, require correction. In our RFI correction, the RFI contamination of the calibration views is only absorbed in the correction of the Earth views.

　　　Finally, as for all intercomparison methods and derived corrections or recalibrations, this RFI correction is only a relative improvement because of the lack of an external reference. Nonetheless, in light of the trend analysis, the stability and consistency of the time series of the various instruments are the most important. These characteristics were improved using our RFI correction and recalibration.

## 6. Conclusions

　　　Analysing inter-satellite biases using global monthly means, we found that a risk of RFI impact exists and may cause biases even if it was not detectable in the verification phase of the instrument. Large biases may emerge from this in the later years of the mission if the gain decreases. Taking the example of NOAA-16 and NOAA-19, we found strong evidence that RFI is the origin for the observed inter-satellite biases compared to NOAA-18. Hence, we found a plausible explanation for these known biases.

　　　Moreover, we provided a correction scheme to mitigate the effects of RFI. This correction scheme operates on the raw Earth counts and corrects these before they enter the calibration procedure. By recalibrating the data for the example instruments on NOAA-16 and NOAA-19, we gained a distinct improvement in terms of the consistency and stability of the brightness temperature time series.

**Author Contributions:** Conceptualization, I.H.; formal analysis, I.H.; investigation, I.H. and M.B.; methodology, I.H.; software, I.H.; supervision, S.A.B.; validation, Imke Hans; writing—original draft, I.H.; writing—review and editing, M.B. and S.A.B.

**Funding:** This research was performed within the project Fidelity and Uncertainty in Climate data records from Earth Observation (FIDUCEO, www.fiduceo.eu) which received funding from the European Union's Horizon 2020 Programme for Research and Innovation, under Grant Agreement no. 638822.

**Acknowledgments:** We also thank Oliver Lemke for his technical support. For providing the level 1b data, we thank the NOAA CLASS.

**Conflicts of Interest:** The authors declare no conflict of interest. The funders had no role in the design of the study; in the collection, analyses, or interpretation of the data; in the writing of the manuscript; or in the decision to publish the results.

## Abbreviations

The following abbreviations are used in this manuscript:

| | |
|---|---|
| AAPP | ATOVS and AVHRR Preprocessing Package |
| AMSU-B | Advanced Microwave Sounding Unit-B |
| APC | Antenna pattern correction |
| ATOVS | Advanced TIROS-N Operational Vertical Sounder |
| AVHRR | Advanced Very High Resolution Radiometer |
| CDR | Climate data record |
| DMSP | Defense Meteorological Satellite Program |
| DSV | Deep space view |
| EUMETSAT | European Organisation for the Exploitation of Meteorological Satellites |
| FCDR | Fundamental climate data record |
| FIDUCEO | Fidelity and uncertainty in climate data records from earth observation |
| FOV | Field of view |
| IWCT | Internal warm calibration target |
| LECT | Local equator crossing time |
| MHS | Microwave Humidity Sounder |
| MW | Microwave |
| NOAA CLASS | NOAA—Comprehensive Large Array-data Stewardship System |
| RFI | Radio frequency interference |
| SSMT-2 | Special Sensor Microwave Water Vapor Profiler |
| UTH | Upper tropospheric humidity |

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
