# Peer review of "Onboard Radio Frequency Interference as the Origin of Inter-Satellite Biases for Microwave Humidity Sounders"

_remotesensing, doi:10.3390/rs11070866_

Round 1

Reviewer 1 Report

In this paper, the strong scan- and time-dependent biases observed for the microwave humidity sounders on board the NOAA-16 and NOAA-19 satellites are analyzed. It is  reported that the radio frequency interference (RFI) is the cause for the biases. The proposed RFI correction scheme increases the stability and consistency between the instruments. Generally, this paper is well written.

The RFI correction method is not explicit. Please consider providing a flowchart to depict the processing procedures. 

Author Response

Response to Reviewer 1 Comments

We would like to thank you for your valuable suggestion which helped to improve the manuscript. In the new version of the manuscript, we implemented the according changes and highlighted these.

Point 1: The RFI correction method is not explicit. Please consider providing a flowchart to depict the processing procedures. 

Response 1: It is a good idea to visualise the steps, thank you for this suggestion. We followed this suggestion and provide a new figure showing a flowchart of the derivation of the RFI-correction. Moreover, we included more detail on the RFI-correction in the text, also adding corresponding equations.

Reviewer 2 Report

In this work, the inter-satellite biases in climate data records from earth observation are studied and connected to the on-board radio frequency interference. A correction scheme also is devised in order to mitigate the effect of RFI.  In general the work presents interest but some points need to be clarified.

1.    There are a number of on-board processing techniques for dealing with the interference in satellite systems (check recommendations below). Is this applicable here, since it might be used to improve the performance.

[R1] C. Politis, S. Maleki, C. Tsinos, S. Chatzinotas and B. Ottersten, "On-board the Satellite Interference Detection with Imperfect Signal Cancellation," IEEE 17thInternational Workshop on Signal Processing Advances in Wireless Communications (SPAWC), Edinburgh, Scotland 2016, pp. 1-5.

[R2] C. Politis, S. Maleki, C. Tsinos, S. Chatzinotas and B. Ottersten, "Harmful Interference Threshold and Energy Detector for On-Board Interference Detection," in 22th Ka Band Conference, Atlanta, USA, 2016.

[R3] C. Politis, S. Maleki, C. Tsinos, S. Chatzinotas and B. Ottersten, "Weak interference detection with signal cancellation in satellite communications," in IEEE International Conference on Acoustics, Speech and Signal Processing (ICASSP), New Orleans, LA, 2017, pp. 6289-6293.

2.    The correction scheme is not properly described. First of all, the novelty with respect the existing works should be addressed. Furthermore, the authors should put more effort to highlight this part of the manuscript since it will strength its technical part. Right now, most of the manuscript seems like a study that is more suitable for a magazine rather than a scientific journal.

3.    A performance metric that can be used to evaluate the effectiveness of the correction scheme will be much more interesting than plotting time series.

4. Revision with respect the Language is needed.

Author Response

Response to Reviewer 2 Comments

We would like to thank you for your valuable comments and suggestions which helped to improve the manuscript. In the new version of the manuscript, we implemented the according changes and highlighted these.

Please find detailed answers below.

Point 1: There are a number of on-board processing techniques for dealing with the interference in satellite systems (check recommendations below). Is this applicable here, since it might be used to improve the performance.

[R1] C. Politis, S. Maleki, C. Tsinos, S. Chatzinotas and B. Ottersten, "On-board the Satellite Interference Detection with Imperfect Signal Cancellation," IEEE 17thInternational Workshop on Signal Processing Advances in Wireless Communications (SPAWC), Edinburgh, Scotland 2016, pp. 1-5.

[R2] C. Politis, S. Maleki, C. Tsinos, S. Chatzinotas and B. Ottersten, "Harmful Interference Threshold and Energy Detector for On-Board Interference Detection," in 22th Ka Band Conference, Atlanta, USA, 2016.

[R3] C. Politis, S. Maleki, C. Tsinos, S. Chatzinotas and B. Ottersten, "Weak interference detection with signal cancellation in satellite communications," in IEEE International Conference on Acoustics, Speech and Signal Processing (ICASSP), New Orleans, LA, 2017, pp. 6289-6293.

Response 1: Since the recommended publications deal with on-board processing techniques that require access to a “living” satellite, as far as we can see, they are not applicable unfortunately. The NOAA-16 satellite died some years ago already, while the NOAA-19 is still alive. However, we are only dealing with data acquired in the past – and also the presented correction can only be derived from existing time series. It is both, a pro and a con: we are able to improve old data, which is helpful from the perspective of a potential data user who wants to study climate.  However, we are (in that sense) only able to improve old data. To correct or even detect the effect of RFI in an operational system, different approaches must be used. For example, during the in-orbit verification phase of NOAA-15, the effect was revealed by switching on and off several transmitters on-board the satellite. However, the effects could not be characterised for each individual pixel (since the scene variability is too large to disentangle it from the disturbing effect). The operational correction of this on-board RFI experienced with the MW humidity sounders is a particular difficulty that will certainly need dedicated efforts. Maybe the techniques that are described in the publications recommended by you, could be investigated in the future in the context of operational correction of the considered MW instrument types.

Point 2: The correction scheme is not properly described. First of all, the novelty with respect the existing works should be addressed. Furthermore, the authors should put more effort to highlight this part of the manuscript since it will strength its technical part. Right now, most of the manuscript seems like a study that is more suitable for a magazine rather than a scientific journal.

Response 2: It is a good point to make the correction scheme more explicit. We extended the description of the correction in the text and we also provide a flowchart to visualise the derivation of the correction. Moreover, we address the differences to the previous correction scheme which was derived during the NOAA-15 in-orbit verification phase and applied to it for some time. To our knowledge, there are no further efforts to reduce the RFI-effect on these MW humidity sounders.

Point 3: A performance metric that can be used to evaluate the effectiveness of the correction scheme will be much more interesting than plotting time series.

Response 3:  Providing a performance metric is an interesting thought to further demonstrate the effectiveness of the correction. Presumably, the performance metric you suggest should provide condensed information on how well the correction works. To measure this effectiveness, different approaches could certainly be thought of. For measuring the effectiveness of the correction the most intuitive way is to investigate how the bias is reduced (which is the overall goal of the correction). We think that this is fairly covered by the time series plots as they reveal the bias in the first place. Also, from a perspective of a user of the MW humidity sounder’s data (to derive higher level products for example), the time series are an important tool, since the time series immediately reveal when the instruments differ from each other – or how biases are reduced by the applied correction. From the time series the user can deduce whether he can build consistent long-term data record or not. Therefore, we would like to keep the time series plots as result.

Point 4: Revision with respect the Language is needed.

Response 4:  We will arrange the required steps with MDPI after the revision.

Reviewer 3 Report

I have reviewed the paper entitled 'On-board radio frequency interference as origin of inter-satellite biases for microwave humidity sounders' by Hans etc.  The paper attributes the biases of microwave humidity sounder on-board NOAA-16 and NOAA-19 to radio frequency interference (RFI). I think there are serious flaws in the analyses.  Fundamentally, one cannot conclude that bias will drift because gain drifts.  It is well known that that gain has been drifting for many satellite channels (e.g., NOAA or EUMETSAT satellite calibration monitoring pages), but inter-satellite biases remain rather stable for many satellite pairs.  I would suggest the authors to first conduct a quantitative correlation analysis between the gain and inter-satellite biases in figure 1 before concluding that the biases were caused by gain changes (last paragraph, page 3).  The relationship between inter-satellite biases and the raw signals as well as the gain are rather complex and I disagree with the statement that 'bias is sensitive to any changes in the raw signals and the corresponding gain'.  The purpose of calibration, although often not perfect, is to make sure biases are not sensitive to changes of raw signals and gain.  Inaccurate calibration non-linearity could be the root cause of the biases, especially for the bias variability in Figure 1.  One should exclude the possibility that inaccurate calibration causes bias changes before attributing them to RFI. 

Because the foundation of the analysis is not solid, it is inappropriate to correct the raw Earth scene counts for temperature bias correction.  There are also other writing problems with the paper.  For instance, the paper is lack of reviews of other bias correction approaches except for RFI.  But because of the fundamental problem with this manuscript, I will not provide detailed review for it.  My recommendation for the manuscript is to reject.                  

Author Response

Response to Reviewer 3 Comments

We would like to thank you for reviewing our manuscript. Your argumentation shows to us that we were not clear enough in some of our explanations. This is a helpful sign and encouraged us to extend and clarify our explanations in the manuscript (changes are highlighted). Please find our detailed response below as “Response 1”.

Point 1: I have reviewed the paper entitled 'On-board radio frequency interference as origin of inter-satellite biases for microwave humidity sounders' by Hans etc.  The paper attributes the biases of microwave humidity sounder on-board NOAA-16 and NOAA-19 to radio frequency interference (RFI). I think there are serious flaws in the analyses.  Fundamentally, one cannot conclude that bias will drift because gain drifts.  It is well known that that gain has been drifting for many satellite channels (e.g., NOAA or EUMETSAT satellite calibration monitoring pages), but inter-satellite biases remain rather stable for many satellite pairs.  I would suggest the authors to first conduct a quantitative correlation analysis between the gain and inter-satellite biases in figure 1 before concluding that the biases were caused by gain changes (last paragraph, page 3).  The relationship between inter-satellite biases and the raw signals as well as the gain are rather complex and I disagree with the statement that 'bias is sensitive to any changes in the raw signals and the corresponding gain'.  The purpose of calibration, although often not perfect, is to make sure biases are not sensitive to changes of raw signals and gain.  Inaccurate calibration non-linearity could be the root cause of the biases, especially for the bias variability in Figure 1.  One should exclude the possibility that inaccurate calibration causes bias changes before attributing them to RFI. 

Because the foundation of the analysis is not solid, it is inappropriate to correct the raw Earth scene counts for temperature bias correction.  There are also other writing problems with the paper.  For instance, the paper is lack of reviews of other bias correction approaches except for RFI.  But because of the fundamental problem with this manuscript, I will not provide detailed review for it.  My recommendation for the manuscript is to reject.                  

Response 1: Unfortunately, we were not able to convince you of our findings in the bias analysis and the deduced correction of the RFI effect. In the following, we refer to your arguments in detail.

Changes in the gain cannot cause a bias by themselves. We agree with you on that. Maybe our wording in the manuscript was not perfectly clear. The important aspect in this statement is “by themselves”: changes in the gain (or the underlying raw signals) do not cause wrong calibration by themselves (they are simply “calibrated out – which is the whole point of the repeated calibration cycles) – in that we agree. But these changes may act as catalyst in the case that additional signals are recorded at the backend of the instrument as it is the case for on-board RFI detected on NOAA-15, and also NOAA-16 and 17 pre-launch (Atkinson2001, ref. [7] in the revised manuscript). In this case, the recorded counts for the Earth views are C’_E=C_E+C_RFI. If the gain and the level of the raw signals C_E are high, the impact of this C_RFI on the final brightness temperature is marginal (this was the case in the early NOAA-16 years). If the gain and the raw signals C_E are very low, however, then each recorded count has a much higher weight. Consequently, the external signal C_RFI (not affected by the decreased C_E and gain, since recorded at the backend only) has a relatively increased impact now. Therefore, only the combination of an extra signal recorded at the backend of the instrument, and a decreasing gain may cause a bias. This is what we observe for both NOAA-16 and NOAA-19 (and NOAA-15 also, for which the RFI-effect is definitively proven in-orbit [7]).

To correct this RFI effect, we modify the Earth counts – note that also the old correction scheme for NOAA-15 in AAPP modifies the Earth counts. However, we provide corrections for each individual FOV (field of view) – whereas the old correction scheme only provided interpolated values between five FOVs. Also, from our retrospective view, we can extract the RFI-zig-zag-pattern better because of more data (compared to the short in-orbit-verification phase).

We agree that other possible sources of the bias (problems in the calibration) should be excluded when assigning the bias to a specific origin. This analysis is undertaken in [10] (which we cite in the introduction), where for the NOAA-16, all other parameters in the measurement equation used for calibration are excluded as possible source for the observed time dependent bias. This is achieved by an analysis of moon-intrusions. Since the analysis did not point to a calibration parameter as source of the bias, the RFI-effect (in combination with the decreasing gain and raw signals, supported by the fact that RFI was measured pre-launch) is the most plausible explanation.

We modified and widely extended the statements and explanations about the relation of the gain, the RFI and the origin of the bias to make clearer the relations and avoid misunderstandings. Moreover, we extended the explanation of the RFI-correction scheme.

Round 2

Reviewer 3 Report

The manuscript has been much improved compared to its original version.  Although I am not completely convinced that it is the RFI that has caused the TB biases in NOAA-16 and NOAA-19; as an assumption, however, the manuscript presents this idea, the correction approach, and its imitations clearly.  I recommend acceptance with minor revision of possible text errors.       

Author Response

Response to Reviewer 3 Comments

Round2

Point 1: The manuscript has been much improved compared to its original version.  Although I am not completely convinced that it is the RFI that has caused the TB biases in NOAA-16 and NOAA-19; as an assumption, however, the manuscript presents this idea, the correction approach, and its imitations clearly.  I recommend acceptance with minor revision of possible text errors.    

Response 1: Thank you very much for revising our manuscript in the second round and recommending acceptance with minor revisions. We will revise the text regarding errors and English language.